# Sorption Features of Polyurethane Foam Functionalized with Salicylate for Chlorpyrifos: Equilibrium, Kinetic Models and Thermodynamic Studies

**DOI:** 10.3390/polym12092036

**Published:** 2020-09-07

**Authors:** Mohamed El Bouraie, Amr Abdelghany

**Affiliations:** 1Central Laboratory for Environmental Quality Monitoring (CLEQM), National Water Research Center (NWRC), El Qanater El Khayria 13621, Egypt; mido.chemie@gmail.com; 2Chemical Engineering Department, Faculty of Engineering, Cairo University, Giza 12613, Egypt

**Keywords:** PUFSalicylate, chlorpyrifos, sorption, Freundlich, Elovich

## Abstract

Commercial polyurethane foam was treated with salicylate salt to synthesize PUFSalicylate (PUFS) as a novel, promising, stable and inexpensive adsorbent for chlorpyrifos (CPF) extraction. The properties of PUFS were investigated using UV–Vis spectroscopy, apparent density, FTIR spectroscopy, pH_ZPC_, BET surface area, elemental analysis, TGA and DTG tests. Batch experiments were performed for the sorption of CPF under different salicylate concentrations, pH, shaking time, ionic strength, agitation speed, sorbent mass, batch factor and initial concentration of CPF. The results confirmed that 97.4% of CPF in a 25 mL solution, 10 μg/L concentration, could be retained by only 0.3 mg of PUFS (1:10^5^mass ratio of PUFS: solution). In addition, the maximum capacity of PUFS for the sorption of CPF was 1249.8 μg/mg (3.9 × 10^−5^ mol/g) within 180 min. The negative values of ΔH° and ΔG° indicated that the sorption of CPF onto PUFS is an exothermic spontaneous process (favorable). The calculated data from the experimental procedures fitted perfectly with Freundlich isotherm (R^2^ = 0.9952) and the kinetic Equation of pseudo-second order. In addition, the R^2^ value in the Elovich Equation recorded higher when compared to the Morris–Weber and Bangham Equations; hence, the pore diffusion is not the rate-determining step. Thus, the use of PUFSalicylate for the removal of chlorpyrifos contaminations from agriculture runoff is applicable.

## 1. Introduction

Pesticides considered priority chemical pollutants in the hazard chemical list due to the extreme toxicity, the effect on vital body functions [1] and the bioaccumulative impact [2,3]. Toxicity of pesticides depended on its chemical characteristic, dose, detention time and exposure pathways [4]. Chlorpyrifos (CPF)—a synthetic organophosphorus pesticide—is used intensively in controlling pests in different crops and soils in developing countries [5,6]. In Egypt, the moderate climate and demand for agricultural productions has resulted in the proliferation of pests and diseases which required the extensive use of CPF [7]. The extensive use of CPF acted negatively on ecological issues, for example, natural disturbance, widespread pest resistance, environmental pollution, hazards to non-target organisms and wildlife and dangers to humans [6,7,8,9].

The chlorpyrifos structure is similar to most synthetic organophosphates (OPs), consistingof three phosphoester linkages termed phosphotriesters and linked to sulfur with a double bond (P=S). Hence, chlorpyrifos remains for a long time in water, soil and sediment according to its strong sorption properties onto suspended organic matter [10,11]. Water contamination with CPF could be widely spread in the natural environment, causing server deterioration to non-target organisms. Furthermore, in the aquatic environment, chlorpyrifos can be transformed into a persistent metabolite called “3,5,6-trichloro-2-pyridinol”, TCP, that is considered as the primary product of chlorpyrifos degradation [12]. TCP has more solubility in water than its primary molecule and causes extensive soil and water contamination due to its resistance to microbial degradation [13,14]. Thus, the development of a novel technique for removal of CPF from the aqueous medium is mandatory.

Several techniques have been applied for elimination of chlorpyrifos from environment (water, soil, plant, etc.) including chemical decomposition, hydrolysis, biodegradation, oxidation, thermal, photodecomposition, ultrasonic treatment and photolysis [15,16]. However, most of the used methods did not address the removal and detention at low concentrations of CPF. Additionally, the commonly used methods were found to be expensive and not always eco-friendly.

Adsorption has been one of the most widely used options for treating organic pollutants in water [17]. Numerous studies have motivated the adsorption process for removing this type of pesticide. Other studies suggested that chlorpyrifos resists biological decomposition because of the accumulation of antimicrobial decomposition products in the environment [18]. Several adsorbent materials have been tested for the sorption of different categories of pesticides from aqueous media, e.g., C18 bonded silica, graphene oxide composites, polymer-coated multi-fibers, nanofibers, activated soil filters, carbon nanotubes, activated carbon and polyurethane foam [19,20,21]. Recently, the use of polyurethane foam (PUF) has increased due to its flexibility, availability, low cost, versatility, high adsorption capacity and efficiency. The presence of polarized and nonpolarized groups within the traditional PUF structure has already been described as the main reason for being able to absorb various inorganic and organic types [22,23,24]. Generally, there are two constraints to use PUF without any chemical pretreatment which are low selectivity for untreatedfoam and congestion of foam surface layer of the adsorbent particles.

Although conventional PUF shows excellent properties, its use for the separation of neutral and acidic organic species is limited because of the basicity of the PUF chain [23]. This limitation necessitates a new type of PUF with a highly acidic character to separate basic organic species. Recently, several studies have shown that the functional primary amino groups of PUF can promote reactivity [25].

In this study, PUF functionalized with salicylate to form the novel and stable PUFSalicylate adsorbent by the coupling between sodium salicylate and the azo group (–N=N–) of the PUF is presented as an eco-friendly technique to optimize the best conditions for the sorption behavior of chlorpyrifos from the aqueous medium. The acidic character of PUFSalicylate (PUFS) makes it suitable to extract some basic organic compounds from wastewater by ionic interactions. PUFSalicylate was compared with PUF in this study and found to be a promising choice for the removal of the chlorpyrifos. Additionally, a relatively small amount of PUFSalicylate was tested and found to adsorb chlorpyrifos efficiently. An experimental batch process has been used for the detection of the chemical and hydrodynamic variables that affect adsorption efficiency. The equilibrium conditions, thermodynamic parameters, and kinetic models have been altered to find the utmost significant adsorption mechanism for the efficient removal of chlorpyrifos from aqueous solution. The main subject of this work is the chemical modification of the white PUF through bonding with salicylate to form a stable chelate, which has high sorption efficiency with the ability to be recycled several times without a significant decrease in their capacities.

## 2. Materials and Methods

### 2.1. Chemicals and Reagents

Chlorpyrifos (C_9_H_11_C_l3_NO_3_PS, O,O-diethyl-O-3,5,6-trichloro-2-pyridyl phosphonothioate) was purchased from Ehrenstorfer, Augsburg, Germany, with 99.0% purification as analytical standard. The chemical structure and characteristics of CPF are listed in Table 1. The stock solution (1.0 mg/L) was prepared using deionized water (Resistance ≈18.2 MΩ cm at 25 °C from 2001-D Siemens System, Vernon Hills, Illinois, USA). Analytical and residue grades of HCl, NaOH, NaNO_2_ and sodium salicylate were purchased from Merck, Darmstadt, Germany. Methylene chloride was used as a chromatograph reagent, and all solvents utilized in the current work were of higher analytical grade (Fluka, Buchs, Switzerland). At the same time, commercial polyurethane foam (white PUF) was purchased from Alfa Foam, Egyptian Co. for Insulation Materials Manufacturing, Cairo, Egypt.

### 2.2. Synthesis of PUFS

The commercial white PUF (10 g) was cut into similarly sized portions, washed several times for one hour by using deionized water, then placed with a 0.1 mol/L HCl solution to remove any impurities. The purified PUF was soaked in 200 mL 6 mol/L HCl for 24 h, the terminal groups of urethane and isocyanate was hydrolyzed after stirring for one hour then placed into a 0.1 mol/L HCl solution and cooled in an ice bath at 4 °C. After this, 50 mL of 2 mol/L NaNO_2_ was decanted gradually under vigorous stirring to the cold solution of the blended PUF until a pale yellow color appeared which indicated the formation of diazonium chloride. For the functionalization step, the mixture was gently boiled with 50 mL 2 mol/L sodium salicylate solution for two hours then left overnight in the fridge. Finally, the pale yellow PUFS formed was washed with distilled water followed by acetone rinsing, then air-dried at room temperature and stored in the dark-glass jar. Hence, the average amount of synthesized PUFS is 11.86 g, as well as its increase in the yield is 15.68%. The proposed pathway of the preparation of the PUFS is displayed in Scheme 1 [26,27].

### 2.3. Apparatus

Spectrophotometric measurements of chlorpyrifos solutions were analyzed using an Orion AquaMate8000 UV–Vis spectrophotometer (Thermo Fisher Scientific, Waltham, Massachusetts, USA) equipped with a 10 mm optical path quartz cuvette. For determining the chlorpyrifos residue, quantitative analysis was performed using the maximum absorption limit at 290 nm.

The modifications and the structural composition of PUFS were investigated by using Fourier-transform infrared spectroscopy (FTIR spectrum 1720, PerkinElmer, Waltham, Massachusetts, USA) in the window of 400–4000 cm^−1^. The morphological properties were magnified by using scanning electron microscope (SEM, JSM 8404, Jeol Limited, Tokyo, Japan). The X-ray diffraction analysis (Rotoflux X-ray diffractometer, XRD, Model 10.61, Bruehl, Germany) was performed with Cu Kα (λ = 1.54060 Å) radiation. The tests were carried out to determine the structural parameters with high precision (20 kV/20 mA and the 2θ scan ranged from 10° to 60° at a step size of 0.04°). Thermal decomposition was performed to examine white PUF and PUFS thermal behavior using a thermogravimetric analyzer (TGA7, PerkinElmer Instruments, Waltham, MA, USA) using nitrogen gas at a heating rate of 10 °C/min from ambient laboratory temperature (20 °C) to 800 °C. The batch experiments were performed utilizing a mechanical shaker (IKA HS 501, Fisher Scientific, Göteborg, Sweden) with a shaking speed in the range between 50 to 250 rpm/min. The pH values were determined using a pH meter (HQ11D Portable, HACH Instruments, Loveland, CO, USA). The specific surface was determined by BET instrument (Micromeritics Gemini 2375 and Gemini V, Unterschleissheim, Germany). The procedure method of the pH drift was carried out to determine the value of pH zero point charge (pH_pzc_). Agilent 7820A gas chromatograph equipped with a nitrogen phosphorus selective detector (GC-NPD) was used for simultaneous identification and quantification of chlorpyrifos. Chromatographic separation was achieved on a CP-SIL 8CB capillary column: length 25 m, column i.d. 0.53 mm and film thickness 1 μm (Agilent Technologies, Inc., Santa Clara, CA, USA). The GC operating conditions were as follows: the injector port temperature was 230 °C and the detector temperature was 270 °C. The oven temperature was 140 °C for 1 min, 140–180 °C at 12 °C/min, 180 °C for 10 min, 180–250 °C at 30 °C/min and isothermal hold at 250 °C for 10 min. Nitrogen was used as the carrier gas at a flow rate of 9.6 mL/min; hydrogen and airflow rate was 3.6 mL/min. The injection volume was 1 μL. Data determination and processing was done using Open LAB CDS ChemStation running on *DELL* computer.

### 2.4. General Procedures

For the sorption of chlorpyrifos, batch experimental procedures were performed based on mixing different amounts of white PUF and PUFS (0.1–0.5 mg/L) with a 25 mL solution containing low CPF concentrations in the range of 1.0–20 μg/L at room temperature. The percentage of CPF residue in solution, initial concentration of CPF, amount of white PUF and PUFS, pH, agitation speed and shaking time were calculated to evaluate removal efficiency of CPF on white PUF and PUFS. Then portions of PUFS loaded with CPF were separated and the residual concentration of CPF in the aqueous solution was quantified at 290 nm using UV–Vis spectrophotometer [26]. After this, the CPF sorption percentage (E%) on PUFS, the capacity of sorbents at equilibrium (q_e_, mg/g) and distribution coefficient (K_d_) were calculated according to the following Equations:E% = [(C_o_ − C_t_)/C_o_] × 100(1)
q_e_ = [(C_o_ − C_e_) × V]/m(2)
K_d_ = q_e_/C_e_(3)
where C_o_, C_e_ and C_t_ are the initial and residual concentrations of CPF in the solution at equilibrium and time (t) during sorption, respectively. V is the nominal volume of solution, and m is the weight of adsorbent. It worth mentioning that all batch experiments were conducted at laboratory temperature with an interval time of 180 min. Additionally, the influence of temperature variation on the thermodynamic parameters and the kinetic measurements was investigated to describe the rate of reaction for the uptake of CPF from water onto the PUFS. Under the equilibrium conditions, calculations were performed to fit the adsorption isotherm model with the experimental data of CPF sorption onto white PUF and PUFS. The solubility of CPF in water was investigated. The results suggested that CPF can dissolve entirely in the water when the concentration is smaller than 1.4 mg/L. As a result, the concentrations of CPF in this work were all smaller than 1.4 mg/L to avoid the possible self-aggregation of chlorpyrifos [11]. Then, the adsorption ability of white PUF and PUFS samples for CPF were compared.

### 2.5. Leaching Tests of the PUFS–CPF

Based on the evaluating solid waste (ESW) and waste extraction test (WET) methods, the leaching test was conducted to evaluate the capability of PUFS as chlorpyrifos storage material [28,29]. The ESW test was carried out by mixing 5 g of PUFS–CPF with 200 mL of the extraction solution (consisted of 0.1 M acetic acid and 0.05 M NaOH) at a pH of 4.93 (liquid/solid ratio of 20:1). The PUFS–CPF and leachate were placed in a capped conical flask (250 mL) and agitated at 60 rpm for 24 h. Once the mixture is filtered (0.45 μm glass fiber filter), the final pH of the filtrate was measured, and the residual chlorpyrifos concentration was analyzed with GC-NPD spectrum. The second test was conducted according to the California WET, by mixing 5 g PUFS–CPF with a leaching agent consists of 0.2M citrate acid at pH = 5 for 48 h (liquid/solid ratio of 10:1). Before the phase separation, the suspension was purged with N_2_ gas for 15 min to release the dissolved oxygen (DO). Successive extractions were performed by mixing PUFS–CPF with different filters under the same preconditions, followed by repeating both tests in six runs with solid-phase separation [30]. It is necessary to measure a blank sample with deionized water before measuring the filtrate. Triplicate samples were measured; hence, the average values and the relative standard deviation percent (RSD%) were calculated. The lixiviated chlorpyrifos from the PUFS–CPF was calculated according to Equation (4):P_CPF_ = C_CPF_ × (L/M)(4)
where P_CPF_ is the CPF leached for a specific liquid/solid ratio (μg/g), C_CPF_ is the concentration of CPF in the leachate (μg/L), L is the volume of suspension (L), and M is the mass of the PUFS–CPF (g).

## 3. Resultsand Discussion

This research focused on reflecting the synthesis, features and applications of polyurethane foam treated with salicylate salt to extract and absorb CPF from the aqueous solution.

### 3.1. Characterization of PUFS

The pale yellow color of the prepared-PUFS was found to be different from the original PUF while maintaining the same cell shape and texture after the functionalization process. The apparent density of the white PUF, PUFS and PUFS–CPF were measured by AccuPycII1345 (Micromeritics Instrument Corp, Norcross, GA, USA) [31] and found to be 28.27, 40.86 and 49.72 kg/m^3^, respectively. The variation in the density of white PUF with PUFS was ascribed to the modification process of PUF by adding cross-linkage bonds between azo groups (–N=N–) in the PUF and salicylate. Figure 1a, described the UV–Vis spectrophotometry analysis, recording that a new low-intensity peak at 286 nm due to the formation of the PUFS chain (the interaction of white PUF with salicylate salt). The analysis ratio of the following elements: carbon, oxygen, hydrogen, nitrogen, phosphorus, sulfur and chloride was summarized as in Table 2. The elemental analysis ratio in three foam molds: white PUF, PUFS and PUFS–CPF indicated a more decrease in carbon, hydrogen and nitrogen in PUFS than white PUF and PUFS–CPF due to the partial hydrolysis of the polymer chain which consists of some groups of isocyanate and urethane associated with CO_2_ released, as well as their incorporation with salicylate salt. Additionally, the oxygen ratio in the PUF is lower than in PUFS and PUFS–CPF due to the presence of the salicylate groups and chlorpyrifos molecules. In comparison, the ratio of phosphorus, sulfur and chloride are recorded only with PUFS–CPF due to the adsorption of CPF molecules onto the PUFS chain.

Brunauer, Emmett and Teller abbreviated as the BET instrument used to determine the specific surface area (m²/g) of PUF, PUFS and PUFS–CPF, including the distribution of pore size shown in Table 2. In general, the BET of the applied materials plays an essential role in indicating the active sites that supported adsorption efficiency. The results indicated that PUFS has surface area of 286.8 m²/g, micropore area of 58.4 m²/g, adsorption pore width of 94.1 Å and desorption pore width of 99.3 Å. The N_2_ adsorption–desorption of PUFS is shown in Figure 1b. Hence, a higher adsorption level was observed at a low relative pressure (P/Po < 0.5). The high formation of micropores enhanced better nitrogen adsorption at low relative pressure. As listed in Table 2, PUFS has a high surface area due to its open porous structure and thus can be used as a template to combine with various compounds as adsorbent and treat aqueous solutions containing organic pollutants. As previously mentioned, the addition of salicylate did not change the image of the porous structure of white PUF; hence, it increased access to more adsorption sites.

The XRD patterns of white PUF, sodium salicylate and PUFS are shown in (Figure 1c) from 2θ = 10–60°, based on all the apparent peaks in this range. Analysis of X-ray Diffraction was used to decide the effect of the intercalation of salicylate on the macro- and microstructure of white PUF. Sodium salicylate crystal had four peaks at 2θ = 20.1°, 23.3°, 27.8° and 29.8°, corresponding to the diffractions of the good crystallinity nature. The sharp and apparent peak of this crystal compound appeared at 29.89°. There was wide diffraction that appeared from 15–25° for white PUF with a maximum peak recorded at 20.85°. For PUFS, the XRD pattern showed similarity to white PUF pattern with the appearance of sharp peaks of sodium salicylate, as in Figure 1c. Introducing sodium salicylate into white PUF resulted in the three characteristic peaks of sodium salicylate overlapping with the white PUF peak as appeared in PUFS and reduced relative intensity after the intercalation of sodium salicylate.

The FTIR spectra of white PUF, PUFS and PUFS–CPF were analyzed using the thin film method. The functional groups of foam molds were specified in the FTIR spectrum, as presented in (Figure 2). For the PUF, the peak was centered at 3000–3500 cm^−1^ due to the stretching vibration of the O–H and N−H group of urethanes, which were confirmed by the appearance of the vibrational band at 1075 cm^−1^ of the –O–CO–NH stretch [22]. The peak band at 1690 cm^−1^ characterizes the C=O stretch indicating the presence of an angular deformation in the N–H plane of urethane. Therefore, the emergence of the band at 2433.5 cm^−1^ is associated with the isocyanate group (–NCO). The appearance of the vibrational band at 1410–1500 cm^−1^ indicates the aromatic C–C stretches, as well as aromatic C–H bonds out of the plane, bend at 754 cm^−1^. However, the FTIR spectrum of PUFS was characterized by another stretching for the broadband of O–H and N–H groups at 3200–3680 cm^−1^, the disappearance of the band corresponding to the isocyanate group (–NCO). Alternatively, new peaks sited at 1260.79, 1438.7 and 1718–1751 cm^−1^ supporting the existence of the OH, N=N and C=O groups were appeared, respectively, after the functionalization. In addition, the existence of a peak at 1100 cm^−1^ corresponds to the C–O and C–O–C groups. Consequently, the disappearance of the absorption peak of the primary amino group (–NH–) in PUF at 1627 cm^−1^ with the addition of a new band at 1570 cm^−1^ for –N=C– to confirm the association of the –N=N– group of the hydrated PUF in the coupling step to form the PUFS. After the adsorption of CPF onto the PUFS chain, The FTIR spectra recorded several new peaks gradually appeared at 635.12, 1050, 1350 and 1480 cm^−1^, corresponding to groups of P=S, S–P–O, P–O–C and aromatic C=C bond. From Figure 2, in the fingerprint area of the FTIR spectra during the sorption of CPF, revealed the peaks of C–Cl with 600–800 cm^−1^ and C–H bonds with 2954 cm^−1^ were ascribed to the presence of CPF onto the PUFS. In particular, the peaks at 1490.06 and 1588.96 cm^−1^ were ascribed to the pyridine ring of CPF. Hence, the P=S vibration peak at 968.12 cm^−1^ was originated from CPF sorption. The peak at 1466 cm^−1^ was assigned to–CH_2_ and–CH_3_ groups, characteristic for the CPF compound. Thus, the potential interaction pathway between CPF and PUFS was confirmed; hence, the success of the adsorption process.

Figure 3 represents the surface morphology of microstructure for three different foam templates as white PUF, PUFS and PUFS–CPF, respectively. Regarding SEM analysis, all images displayed the microstructure of open-cell chains as dodecahedral cavities with categorized according to the distribution and linkage of internal pore strain. It is reflected in strong binding between open dodecahedral holes, as observed in Figure 3a, then gradually decreased with the addition of salicylate and adsorption of CPF molecules. For Figure 3b, the distribution of smaller pore diameters, indicating that the image of loaded foam has higher homogeneous morphology. As already displayed, the PUFS synthesized in the current study has a well-enhanced open-cell structure with a non-regular distribution of salicylates. The SEM image of PUFS–CPF at high magnification showed the incorporation of CPF with PUFS, as in Figure 3c. Hence, the distribution of CPF molecules on the PUFS is ascribed to enhance the microstructure of PUF through the functionalization with the salicylate. As a result, it is considered one of the appropriate adsorbents for extracting different pollutants from wastewater.

The pH_zpc_ values were varied among the PUF and PUFS to be approximately 8 and 6, as in Figure 4a. Basically, at the pH_zpc_ value, there was no surface charge to be neutralized form due to the existence of a perfect charge balance between the equilibrated ions in an aqueous solution. At pH lower than the pH_zpc_ value, the surface became positively charged, whereas, at pH higher than pH_zpc_, the surface became negatively charged [32]. The pH_zpc_ value was based on the functional groups that were most likely membrane structures such as NHCOO and terminal OH groups. The result indicated that the pH_zpc_ of the PUF was higher than PUFS due to the existence of the carboxylic functional group in salicylate salt which decreased it to 6.

TGA and DTG tests were conducted on white PUF and PUFS to evaluate the influence of different temperatures on the thermal decomposition. For Figure 4b, the TGA plot was performed to evaluate the thermal deterioration stages for both foam molds according to the percent of weight loss. In the beginning, the weight of PUFS was almost stable more than white PUF between 120 and 180 °C. It can be seen that the deterioration of both samples emerged in two stages; therefore, the first stage of PUF initiated at 120 °C to decrease gradually up to 240 °C, the weight loss ratio was 32.7%, ascribed to the breakdown of cell structure to isocyanate, urethane and dissociation of the urethane bond to form amine derivatives. The second stage extended between (240–380 °C), the ratio of weight loss decreased significantly (57.5%), due to the dissociation of remaining fragments from the first stage with the release of CO_2_. For higher than 380 °C, the ratio of weight loss recorded a constant rate of about 11.6%.

In contrast, the thermal deterioration of PUFS starting gradually at higher than 285 °C, while increased significantly at more than 320 °C to record the following mass loss ratios (22.34%, 66.7% and 4.2%) in the ranges of 180–265, 265–390 and more than 390 °C, respectively. Therefore, the cell structure of PUFS was more stable (less elasticity) than that of PUF at high temperature due to the attraction bond between–NCO of PUF and the–COO of salicylate was also useful to the thermal stability. Moreover, the DTG plot of PUFS revealed two endothermic peaks at 335.4 and 380.68 °C as in Figure 4c.

### 3.2. The Optimum Condition for Sorption of Chlorpyrifos

#### 3.2.1. Effects of Salicylate Concentration on the Functionalization Process

According to the type and quantity of functional groups added to the pure polyurethane foam can specify the adsorption capacity and the optimal dose. Therefore, several functionalized samples were synthesized by adding different concentrations of sodium salicylate in the range (0.5–5.0 mol/L) to test the adsorption efficiency of CPF from aqueous solutions. For Figure 5a, the result indicated that any increase in the uptake of CPF is ascribed to the concentration added of salicylate salt onto PUF. Hence, the optimum concentration of salicylate salt was identified at 2 Mandapplied in the following experiments due to the convergence when concentrations were higher than 2 mol/L [26].

#### 3.2.2. Effects of PUFS Mass

Polyurethane is an organic polymer widely used as a conveyor. However, it contains many physicochemical aspects like small density, large area, hydrophobic and high absorbency. According to the earlier results, the adsorption procedure depends on the ion association between the cationic salicylate and anionic pesticide, chlorpyrifos, where CPF is slightly soluble in water below 1.4 mg/L [11,33]. It is necessary to functionalize the hydrophobic PUF material in order to enhance the CPF retention ability and avoid potential self-aggregation. Furthermore, using the optimum concentration of salicylate to the PUF was an essential stage to promote CPF adsorption by varying the dosage of PUFS in the range (0.1–0.5 mg/L) under experimental conditions. Figure 5b displayed the effect of the PUFS dose on the sorption efficiency of CPF in aqueous solution. The variation in the dosage of PUFS employed was not observed when a higher amount than 0.3 mg/L was added, probably due to the increased blocking of active sites from the aggregation of CPF molecules, as shown in Figure 3c. Hence, the dominant dose of PUFS, 0.3 mg/L was detected for the next experimental procedures for the extraction process at 30 min. The role of the PUFS dose was evaluated through its effect on the mechanism and the effectiveness of active sites.

#### 3.2.3. Effects of pH, Shaking Time and Agitation Speed

The batch system was used to investigate the effect of pH on the CPF sorption onto the white PUF and PUFS. In practical, sequential experiments were conducted to monitor the pH effect on different CPF concentrations from 1.0 to 20.0 μg/L, similar to a sample of agricultural wastewater effluent. Varying the amounts of the adsorbent for each of the foam molds (0.1–0.5 mg/L) was tested under certain conditions of stirring and temperature with time intervals up to 180 min. Variations in CPF sorption with pH were investigated by adapting the foam suspension pH with 0.01 mol/L NaOH and HCl. The results indicate that the maximum sorption of CPF onto white PUF occurred in the pH range of 2–7 while the maximum sorption occurred in the pH range of 2–5 for PUFS (Figure 6a). Furthermore, the optimum pH value to remove CPF from aqueous solution was 3 and 4 for the white PUF and PUFS, respectively. At pH = 2, high sorption efficiency was observed due to the consistency of a positively charged onto the surface of each foam molds to enhance ion-association with CPF molecules that were ascribed to the protonation of all nitrogen atoms in the urethane linkage [34]. In contrast, at pH value more than 7.0, the sorption percentage of CPF for both foams molds is small, and then it decreases with the increase of pH. The result clarified that the sorption of CPF under alkali condition decreased due to the decomposition of CPF molecules to form 3,5,6-trichloro-2-pyridoinol, TCP, as chlorpyrifos byproduct. Figure 6b summarizes the sorption percentage of CPF after 180 min in the order of PUFS (97.4%) > white PUF (35.2%); this depended on the surface charge, ion association, adsorbent surface characteristics and size of CPF molecule. Hence, the curve of white PUF could reasonably suggest that the maximum sorption percentage of CPF may occur as surface sorptionaccording to the electrostatic attraction mechanism with CPF [32,34]. At the same time, the sorption percent of CPF by PUFS recorded a significant increase compared to the white PUF due to the protonation of oxygen functional groups on the PUFS surface. Oxygen functional groups appeared on the peak corresponding to C–O and C–O–C groups at 1100 cm^−1^, as in Figure 2, to interact with the nitrogen atoms and pyridine ring of chlorpyrifos at low pH value [8,34]. In addition, the pH_ZPC_ of the PUFS surface is about 6, which is suitable for adsorption CPF (negatively charged), while the size of the aromatic ring of CPF is less than the average micropore surface area of holes inside PUFS (58.4 m^2^/g).

The procedures of the sorption between chlorpyrifos from the bulk solution and interface of the two solid-phases were based on the shaking time and agitation speed. Hence, the shaking time can impact on the sorption percent to regulate the competence order of white PUF and PUFS action towards CPF. At the same time, the shaking speed of the species increased the collision rate between the active sites in the liquid–solid phase and increased the average kinetic energy in the system. Therefore, agitation speed and shaking time were investigated in the range of 20–90 rpm and 0–180 min, respectively. Those experiments were conducted using a bath shaker with 0.3 mg of white PUF or PUFS and initial CPF concentration was 2.85 × 10^−5^ mmol/L (10 μg/L) at 25 °C. For evaluating the shaking time, the results revealed that the maximum sorption efficiency of CPF onto white PUF and PUFS is completed within about 60 and 30 min, respectively. According to the results concluded in Figure 6b, the adsorption equilibrium between CPF and PUFS is faster and more efficient than that of white PUF.

In contrast, the agitation speed did not affect the sorption efficiency of CPF on the PUFS ultimately. In this context, a sorption percentage of approximately 97.2% was observed in the whole range studied, as in Figure 6c. Furthermore, the results indicated the shaking time of white PUF for 60 min or more recorded the better sorption of CPF from the aqueous medium, indicating that a maximum agitation speed of 100 rpm is required to assure the maximum collision of CPF from solution to the white PUF surface. This result implied that white PUF retention is relatively slow.

#### 3.2.4. Effects of the Ionic Strength and Batch Factor

The sorption of CPF molecules was conducted under the influence of ionic strength. The batch experiments were performed under optimum conditions by using different concentrations of individual electrolytes (0.5, 1.0, 3.0 and 5.0 mol/L KCl, NaCl and NH_4_Cl). Figure 7a shows that sorption percent decreased with an increase in ionic strength of the salt solutions for the two foam molds. The decrease in sorption percent was ascribed to the attraction between the adsorbent surface and Cl^−^ ions. The low sorption of CPF molecules at a high concentration of electrolytes was attributed to the competition between chloride ions with anionic pesticide molecules affected by the size of adsorbate and the strength of attraction bond. At the same time, a considerable increase in the electrolytes led to a decrease in the sorption [24]. The low in the sorption of CPF corresponding to raising ionic strength was ascribed to first, increasing the concentration of K, Na or NH_4_ that lead to increase competition on the sorbent sites. Second, decreasing CPF activity due to the formation of positive and negative electrostatic shell around foam and pesticide molecules, respectively, a result of the increase in the concentration of the electrolytes in the solution. This behavior clarified the shrunk of the dodecahedral pores in the foam structure at high ionic strength and effectively decreased the pore size of the cavity structure. The existence of the Cl^−^ ions in high concentrations at the foam surface promotes the internal pore strain then impairs the surface potential more rapidly. In contrast, the relative change in the extraction of CPF at lower ionic strength for two foam molds was ascribed to the behavior of ion-exchange effects enhanced after the functionalization of the PUF by salicylate.

The solid-phase sorption system was used for the sorption and extraction of pesticides from large volumes of the samples [35]. The performance of white PUF and PUFS in the preconcentration of chlorpyrifos in different volumes of the aqueous solution was investigated. The effect of the sample volume (25–150 mL) to the weight of white PUF and PUFS (0.2 mg/L) on the sorption percentage (%) of CPF (10 μg/L) was performed using batch technique. The data suggested that the maximum CPF sorption in batch factor (V/m) ranged from 50 to 125 for white PUF ranges and from 50 to 250 for PUFS. The sorption of CPF decreased with increasing of the sample volume (34.6% to 31.8%) and (97.4% to 95.5%) in batch factor, V/m ≈ 220 and 350 for white PUF and PUFS, respectively (Figure 7b). Moreover, the effect of sample volume (25 mL) on different amounts of white PUF and PUFS (0.1–0.5 mg) was investigated. The obtained results observe that the maximum CPF sorption percentage noticed in (0.4–0.5 mg) and (0.3–0.5 mg) in batch factor, (V/m ≈ 100–250) of white PUF and PUFS, respectively.

#### 3.2.5. Effects of Initial Concentration

The effect of initial chlorpyrifos concentration on the sorption capacity with white PUF and PUFS was investigated in the range of 1.0–20.0 μg/L (2.85 × 10^−6^–5.7 ×10^−5^ mmol/L). Figure 8 depicted the linear plot between the sorption capacity of CPF against different initial concentrations, where the value of R^2^ was 0.8255 and 0.9912 for white PUF and PUFS, respectively. Whereas the sorption capacity of PUFS increased rapidly with increasing CPF concentration and then slowly progressed to equilibrium over time ([CPF]_O_: 15 μg/L). Rapid initial extraction may be ascribed to the attraction of CPF molecules with available surface active sites while following gradual sorption may be interpreted to proton displacement and chelation between the adsorbent pores with functional groups of CPF molecules. Regarding that, the maximum CPF sorption capacities onto white PUF and PUFS were determined to be 312.5 and 1249.8 μg/mg, respectively. Hence, the sorption capacity of PUFS is relatively better than white PUF. For Figure 5b, the equilibrium was achieved for the sorption of CPF at 30 min with PUFS of (0.1–1.0 mg/L) and at 45 min of (1.5–2.0 mg/L). In the same context, for white PUF, the equilibrium was recorded at 60 min with all initial concentrations.

Consequently, the distribution coefficient (K_d_) values for CPF molecules were 40.12 and 563.67 L/mg on white PUF and PUFS, respectively, as in Figure 5b. These results confirmed the limitation of active sites on the white PUF surface. In addition, at higher concentrations of pesticide, large doses of adsorbent were required to complete the sorption process due to saturation of currently active sites. In contrast, the efficiency of PUFS has been demonstrated as a polymeric extractor to enhance the sorption of CPF molecules to become similar to the liquid/liquid extraction system [35,36]. A comparison of the sorption capacity of the prepared PUFS with other adsorbents denoted that it has a promising efficiency for chlorpyrifos sorption, as listed in Table 3.

#### 3.2.6. Effect of Temperature and Thermodynamic Study

The sorption percentage of CPF onto white PUF and PUFS were investigated under the equilibrium status as a function of temperature (20–60 °C). The obtained results show that there was a reverse effect of temperature on the sorption of CPF onto the white PUF, as depicted in Figure 9a. The best result of the CPF sorption on white PUF was recorded at lower temperature values. In contrast, the CPF sorption on PUFS slightly decreased with increasing the temperature due to the existence of salicylate groups, which led to lower form temperature by dilating superficial dodecahedral pores of the PUFS. Hence, the resultant decrease in CPF sorption was attributed to the disintegration of bonds between active adsorption sites and the CPF molecules. The thermodynamics for the CPF sorption followed equations:lnK_d_ = −(∆H°/RT) + (∆S°/R)(5)
∆G° = ∆H° − T∆S°(6)
∆H° = E_a_ − RT(7)
where K_d_ is the distribution coefficient for CPF sorption, ΔH° is the enthalpy change, ΔG° is the Gibbs energy change, ΔS° is the entropy change, E_a_ is the activation energy, R is the universal gas constant and T is the absolute temperature in Kelvin. All the thermodynamic parameters can be obtained from estimating the slope and the intercept values of the linear plot of lnK_d_ against 1/T. Figure 6b indicated a linear relationship between lnK_d_ and 1/T for CPF sorption onto white PUF (R^2^ = 0.9461) and PUFS (R^2^ = 0.9973) under equilibrium conditions in the batch system. Table 4 summarizes the various thermodynamic parameters deduced from the above plot (Figure 6b). The average calculated free energy (ΔG°) values were −8.85 and −19.97 KJ/mol onto white PUF and PUFS, respectively, where the negative values were ascribed to the spontaneous behavior of the current adsorption process. The results recorded increased negative ΔG° values with increased temperature to confirm the spontaneity of CPF sorption onto PUFS. In addition, the values of ΔH° for CPF sorption onto white PUF and PUFS are −20.8 and −13.1 KJ/mol, respectively. In general, the negative values further suggesting that the sorption process is exothermic chemisorption [40]. However, the negative values of ΔS° (for PUF, −38.2 JKmol, less disorder) reflects that no significant change occurs in the internal structures of the PUF during the sorption process. In contrast, the positive value of ΔS° (for PUFS, 21.9 JKmol, more disorder) can be interpreted as the relatively fast CPF adsorption rate due to the excess energy generated by randomness between the two interfaces in the batch process. Regarding the results, the CPF sorption process onto PUFS was faster than PUF due to the small values of E_a_ and ΔH° as well as the driving force like proton displacement and chelation between functional groups of PUFS and pyridine ring and the amine group of CPF molecule [13,34,41].

### 3.3. Sorption Isotherm

The sorption process was investigated through sorption isotherm to reveal a suitable model for simulating the sorption approach. The following models of isotherms were used to describe the quantitative interaction between solutes and adsorbents. Under optimum conditions, sorption isotherms provide more details about the nature of the sorption process in the batch system. The data obtained were examined according to four adsorption isotherms, namely, Langmuir, Freundlich, Dubinin-Radushkevich (D–R) and Temkin models for the sorption of CPF on PUF and PUFS. The Langmuir model was selected to appreciate maximum adsorption capacity correlated with monolayer coverage on the adsorbent surface and is given by [42]:C_e_/q_e_ = 1/K_L_q_max_ + C_e_/q_max_(8)
where K_L_ is a constant correlated with adsorption/desorption energy (L/g), C_e_ is the concentration of CPF at equilibrium (μg/L), q_e_ is the amount of CPF adsorbed per unit mass of adsorbent (μg/g) at equilibrium and q_max_ is maximum sorption of the full saturation of the adsorbent surface (μg/g). The experimental data were represented by the linear plot of C_e_/q_e_ versus C_e_, as in Figure 10a. The Freundlich model was identified to decide the adsorption density of chlorpyrifos not only towards uniform adsorption, but to include multilayer and was determined by [43]:logq_e_ = logK_f_ + 1/n (logC_e_)(9)
where K_f_ and n are the Freundlich constants. The value of n suggests the affinity of CPF towards the adsorbent. The linear plot of log q_e_ versus log C_e_ for the Freundlich isotherm as in Figure 10b. Dubinin-Radushkevich isotherm was selected to determine the adsorbent porosity and the evident energy of adsorption. The D–R model is expressed by [44]:lnq_e_ = lnq_d_ + βε^2^(10)
where q_d_ is the D–R isotherm constant correlated with the degree of CPF sorption by the adsorbent surface; ε is the Polanyipotential calculated by [ε = RTIn(1 + 1/C_e_)], β is associated with apparent energy of adsorption per mole energy (E_a_) of CPF emigrated to the adsorbent surface from infinite distance in the suspension (mol^2^/kJ^2^); hence, free energy calculated by [E_a_(KJ/mol) = (2β)^−0.5^], R is the gas constant (8.314 J/mol K), and T is the temperature of the solution (K). The D–R isotherm plot of lnq_e_ against ε^2^, as in Figure 10c. The Temkin isotherm supposed that the fall in the adsorption heat was linear instead of logarithmic, as estimated in the Freundlich Equation [45]. For Temkin isotherm, the linear plot of q_e_ versus lnC_e_ is plotted according to the Equation:q_e_ = (RT/b_T_) lnK_T_ + (RT/b_T_) lnC_e_(11)
where K_T_ (L/mg), is the equilibrium constant that correlated with the maximum linkage energy and b_T_ (kJ/mg), is the variation of adsorption energy that is regarding the adsorption heat (ΔH°_ad_). Temkin isotherm plot is displayed in Figure 10d.

Table 5 lists the constant values of the employed isotherms were estimated from the slope and intercepts of the various linear plots and corresponding values of the linear regression coefficient (R^2^). The results indicated that maximum uniform coverage for the Langmuir model (q_max_) was 1249.8 μg/g of PUFS and 312.5 μg/g of white PUF. Calculated R_L_ values were less than unity while the value of n for PUFS was slightly lower than unity; hence, the sorption of CPF onto PUFS was appropriate. Since 1/n value is close up unity (1/n = 1.085), the adsorption efficiency becomes more feasible with lower concentrations of adsorbate, as well as is more effective at higher concentrations and vice versa. For PUFS, the linear plot of the Freundlich model yielded the best R^2^ value for CPF sorption (R^2^ = 0.9952), which was higher when compared to white PUF [43]. Based on the heterogeneous structure of PUFS, the adsorption process was not onto a monolayer surface rather multilayer sorption performed in the PUFS–CPF system. Moreover, monolayer adsorption also represented a significant role in CPF extraction onto PUFS (R^2^ value for Langmuir is 0.9893). The D–R model constants indicated that q_d_ value was highest for PUFS and lowest for white PUF. Regarding the q_d_ value, the obtained results for white PUF and PUFS were somewhat compatible with the qmax value than estimated for the Langmuir isotherm. The values of R^2^ for the D–R isotherm were lower than Freundlich and Langmuir isotherm models, as listed in Table 5. The apparent energy of adsorption E_a_ (KJ/mol) gave 12.7 for PUFS and 3.98 for PUF. The calculated E_a_ values can be employed to recognize the reaction mechanism; hence, E_a_ value of 8–16 kJ/mol demonstrates an ion exchange or chemical reaction processes, however, in a physical adsorption E_a_ value < 8.0 kJ/mol [46]. Therefore, the obtained result of E_a_ value for PUFS confirmed that the sorption processes followed the chemisorption mechanism while for white PUFwas physical adsorption due to E_a_ value of 3.53 kJ/mol. The variation in E_a_ values between PUFS and white PUF suggested that the rate of adsorption onto PUFS was relatively faster. Based on the results obtained from the linear plot for Temkin isotherm (R^2^ ≥ 0.97), which appreciated the chemisorption of CPF onto PUFS. Consequently, this result indicated that CPF sorption onto white PUF was a physical process, but PUFS was a chemisorption process [47]. The adsorption isotherms sequence for CPF onto PUFS was in order Freundlich> Langmuir >Temkin > D–R isotherm while for white PUF was Langmuir > D–R > Freundlich > Temkin isotherm. Besides the hydrophobic attraction, other mechanisms as proton displacement and chelation performed a vital role in the sorption of chlorpyrifos onto PUFS. Chlorpyrifos molecule has functional groups of amine and pyridine ring that contributed to confirm the chemisorption mechanism on PUFS besides hydrophobic attraction and surface adsorption onto white PUF.

### 3.4. Sorption Kinetics and Intraparticle Diffusion

The experimental kinetic data were investigated by three kinetic Equations, namely: pseudo-first-order, pseudo-second-order and Ritchie’s second-order Equations, while the intraparticle diffusion was evaluated by four intraparticle diffusion Equations, namely; Morris–Weber, Bangham, Elovich and Reichenberg Equations. Kinetic models play an important role in confirming the potential mechanism of chlorpyrifos adsorption and controlling the rate of the adsorption process. Hence, the kinetic Equations are expressed as follows; Pseudo-first order [42];
ln(q_e_ − q_t_) = lnq_e_ − K_1_t(12)

Pseudo-second-order [42],
t/q_t_ = 1/K_2_q_e_^2^ + t/q_e_(13)

and Ritchie’s second order [48].
1/q_t_ = 1/Kq_e_t + 1/q_e_(14)
where q_e_ and q_t_ (μg/g) are the amounts of CPF adsorbed at equilibrium and time t (min), respectively; K_1_ is the rate constant for pseudo-first-order Equation; K_2_ is the rate constant for pseudo-second-order Equation. h_o_ is the initial sorption rate for pseudo-second-order Equation (h_o_ = K_2_q_e_^2^) and K is the rate constant for Ritchie’s second-order Equation. Figure 11a–c displayed the linear plots of three kinetic models to estimate the constant parameters from the slope and intercept under the optimum condition for CPF sorption onto white PUF and PUFS. The calculated parameters of the studied kinetic models arelisted in Table 6. As in Figure 11a, CPF sorption linear plot on white PUF indicates that the pseudo-first-order process was divided into three phases, while the linear plot of PUFS appeared in one phase and more consistent [23,36]. The mean values of rate constant (K_1_) of CPF sorption onto white PUF and PUFS estimated from the slopes were 0.31 and 1.26 min^−1^, respectively. The calculated values of the half-life of CPF sorption (t^1/2^) were 2.24 and 0.55 min (Table 6). Based on the low values of R^2^, pseudo-first-order has not demonstrated to be favorable for CPF absorption. For pseudo-second-order, the mean values of rate constant (K_2_) of CPF sorption onto white PUF and PUFS derived from the slopes were 0.46 and 2.8 min^−1^, respectively. The estimated values of the half-life of CPF sorption (t^1/2^) were 2.17 and 0.36 min (Figure 11b). Ritchie’s’ second-order Equation recorded moderate rate constant (K_R_) values for CPF sorption onto white PUF and PUFS while the values of R^2^ were lower than for the corresponding for pseudo-second-order as in Figure 11c. High R^2^ value distinguished the process of pseudo-second-order (0.9981), indicating that this kinetic model was more dominant in the sorption of CPF, as in Table 6. These results suggest that the rate constant of the adsorption based mainly on the reaction of pseudo-second-order between functional groups of PUFS (chelating groups) with the amino group or pyridine ring of CPF molecule within a short period [36,42].

Generally, all the phases of adsorption (internal particle diffusion, external film diffusion and surface adsorption) fall within pseudo-second-order kinetic. Hence, it was important to detect the rate-controlling step of the sorption system. The intraparticle diffusion Equations investigated in this work are expressed as follows; Morris–Weber model [42];
q_t_ = K_i_t^1/2^ + C(15)

Bangham Equation [49],
log log(Co/Co− Mq_t_) = log(K_B_M/2.303V) + αlogt(16)
and Elovich Equation [50].
q_t_ = 1/βln(αβ) + 1/βlnt(17)
where q_t_ (μg/g) is the amount of CPF adsorbed at time t; K_i_ (mg/g min^1/2^) is the rate constant for intraparticle diffusion by Morris–Weber; C is the intercept value to give a typical point of view on the thickness of the boundary layer. α_e_ and β_e_ are Elovich constants that correlated to chemisorption rate and the extent of surface coverage; V is the volume of CPF solution in mL; M is the mass of adsorbent (g/L); α and K_B_ are Bangham constants to appreciate if pore diffusion is a typical step to control the adsorption rate. Figure 12a–c displayed the linear plots of three intraparticle diffusion Equations to decide the constant parameters from the slope and intercept under the optimum condition for CPF sorption onto white PUF and PUFS. The estimated parameters of the studied kinetic models were recorded in Table 7. Morris–Weber Model was used to recognize the potential of the internal diffusion rate (K_i_) of the sorption of CPF onto white PUF and PUFS. Figure 12a revealed that the linear regression of the plot did not pass through the origin (The value of C is not equal to zero) as well as C value was too far-well than zero for PUFS more than white PUF. Therefore, internal diffusion has not become the rate-limiting step [24]. As in Figure 12a, the diffusion rate was rapid in the early stages, and the values of diffusion constant (K_i_) were 1.77 and 1.66 μg/g min^1/2^ for CPF onto white PUF and PUFS, respectively. For the Bangham model, the decided results of α and K_B_ constants significantly reveal that multiple adsorption procedures occurred as in Table 7. The values of R^2^ obtained from Morris–Weber and Bangham models were 0.9493 and 0.962, indicating that pore diffusion was concerned in CPF sorption onto PUFS. However, the linear plot of the Bangham Equation and the value of α < 1 confirmed that sorption pore diffusion was not the rate-limiting step (Figure 12b) [49]. β_e_ is a constant correlated with the sorption of surface coverage and decreased with increased initial CPF concentration (Figure 12c); hence sorption of CPF onto PUFS may be occurred more typically by functional groups. Whereas α_e_ was associated with chemisorption rate that has increased with increasing initial CPF concentration, this demonstrates that different pathways control CPF sorption onto PUFS [50]. For Table 7, the Elovich Equation recorded higher R^2^ values more than Morris–Weber and Bangham Equations; hence, it became more favorable to the sorption process [23,49].

### 3.5. Leaching Test Results

To confirm the capability of the synthesized PUFS, not only as a novel adsorbent, but alsoas a pollutant confinement material, leaching tests were conducted by mixing PUFS–CPF with different agents in the batch system according to the ESW and WET. The obtained results of the ESW and WET are shown in Table 8. Figure 13 displayedthe leaching percent of CPF in each run of two consecutive extraction procedures. It can be seen that the leached CPF was below the detection limit (0.03 μg/L) after Steps 1 and 2 in both tests. Hence, even under conditions of ESW and WET tests and with repetition procedures up to six times, it was found to leachate 3.5%–4.2% of chlorpyrifos retained in PUFS, respectively. In both tests, agitation assay conditions were performed to leach CPF, which confirms that PUFS could be a good selection as a pesticide confinement material. It is necessary to observe that none of the diluted solutions of acetic or citric acids could leachate more than 10% of CPF retained in the PUFS, similarly to the small loss by leaching chlorpyrifos from sediment was 0.003% due to mainly associated with carried sediment [11,13].

### 3.6. Economic Study

Polyurethane foam was spread in the surrounding environment as waste at no cost; moreover, the real cost of collection and processing to reuse as a typical adsorbent would equalize about 30 USD/ton. Comparing the eco-feasibility study of PUFS with other adsorbents for water treatment, the chemical modification of available adsorbent may lead to a cost increase of about 10 times while commercial activated carbon is about 500 times more expensive than PUFS. Recently, large amounts of environmental waste from polyurethane compounds have been observed in their various forms due to their low cost and ease of use. In developing countries, people may usually use these as utensils of takeaway food. The proposed method is a safe alternative for disposing of foam products. Thus, PUFS gives a useful economic feature over commercial activated carbon and other modified adsorbents.

## 4. Conclusions

The synthesis and characterization of a new solid-phase based on the incorporation of salicylate with commercial polyurethane foam were performed. Salicylate group was incorporated through (–N=N–) group in order to eliminate the problem of ligand leaching. The sorption behavior of chlorpyrifos (CPF) onto PUFSalicylate (PUFS) was investigated to decide the best conditions for its removal and preconcentration from aqueous solution. Under optimum conditions, the maximum capacity of PUFS was 1249.8 μg/mg (3.9 × 10^−5^ mol/g) to retain the solute within 180 min. The thermodynamic parameters suggest that sorption of CPF onto PUFS is spontaneous and exothermic. Adsorption isotherm was determined to describe the nature of the sorption process by using four isotherms: Langmuir, Freundlich, Temkin and Dubinin-Radushkevich (D–R) models. Hence, the equilibrium conditions of the liquid/solid system were favorably depicted by the Freundlich isotherm for CPF onto PUFS. The kinetic Equation of pseudo-second-order a well-fitted for describing the sorption of CPF onto PUFS. In addition, the sorption rate of CPF onto PUFS was 2.8 min^−1^, with the half-life of sorption (t^1/2^) at 0.36 min. Based on the results of Morris–Weber and Bangham models as well as the Elovich Equation, the system sorption was a chemisorption type, and pore diffusion was not the rate-controlling step (as ion-associates between the amine group or pyridine ring of chlorpyrifos with PUFSalicylate). Moreover, both leaching tests confirmed that PUFS could be a good selection as chlorpyrifos confinement material. The proposed sorbent is a promising material when applied for removing chlorpyrifos from agriculture runoff.

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
