# Peer review of "Sorption Features of Polyurethane Foam Functionalized with Salicylate for Chlorpyrifos: Equilibrium, Kinetic Models and Thermodynamic Studies"

_polymers, 2020, doi:10.3390/polym12092036_

Round 1

Reviewer 1 Report

PU foam is very useful in different applications. Authors functionalized PU foam to act as an adsorbent. The following points need to consider to improve the manuscript

  1. In Figure 2, the spectra has not been marked! 
  2. FT-IR explanation need revision, few sentences are confusing! The peak for NCO group is not detectable in Figure, 
  3. Please cite reference to support your synthesis 
  4. Total 68 reference! many!
  5. The manuscript need revision by a native English speaker, 

Author Response

Reply of Authors to Reviewer 1

I want to thank the reviewer for his/her time, observations, and comments. The comments helped us to improve our work so that it can be easily accessible to readers in the scientific community.

Comment 1. In Figure 2, the spectra has not been marked! 

Authors’ Response: Thanks for highlighting this critical point. The spectra were marked in details in the revised version of the manuscript

Comment 2. FT-IR explanation need revision, few sentences are confusing! The peak for NCO group is not detectable in Figure, 

Authors’ Response:Thanks for addressing that issue. This point was well taken, and FT-IR was explained and highlighted in the revised version of the manuscript. Additionally, the peak for the NCO group and other groups were outlined in the modified figure 2.

Comment 3. Please cite reference to support your synthesis 

Authors’ Response:A reference was cited to support synthesis, as advised. Thank you!

Comment 4. Total 68 reference! many!

Authors’ Response:Thanks for the comment. We reduced the number of references in the revised manuscript.

Comment 5. The manuscript need revision by a native English speaker

Authors’ Response:Thanks for the comment. The abstract and manuscript were revisited, and a commercial, technical writing software was used to address and correct mistakes in the English of the manuscript.

Thank you, we hope this revision has been enough.

Reviewer 2 Report

In this manuscript, the authors reported the functionalization of PUF with salicylate to form PUFS, which could be used as an excellent adsorbent for extracting chlorpyrifos (CPF) from aqueous solution. For this aim, first the structure and properties of the formed PUFS were characterized with multi-techniques. In addition, the adsorption experiments indicated that PUFS exhibited very good adsorption capacity towards CPF with a maximum capacity of 1249.8 μg/mg within 190 min. therefore, this designed PUFS has great potential for environmental remediation. It is an interesting work. The experiments are good-designed, and the manuscript is well-written and good-organized. All the obtained results are convincible, which provide good supports for the presented conclusions. Based on these points, this manuscript is recommended for publication at Polymers after minor revisions.

Special comment for the revision:

  1. In the “Introduction” part, it is suggested for the authors to give more information on the novelty and significance of this manuscript.
  2. It is suggested for the authors to add a scheme to present the synthesis of PUF and the PUFS, as well as the adsorption of CPF.
  3. In Figure 8a and 9b, the error bars of data points should be added.
  4. To prove the advantages of the formed PUFS for removing CPF by comparing with other materials, the authors should add the comparison on the adsorption performance of all materials.

Author Response

Reply of Authors to Reviewer 2

I want to thank the reviewer for his/her time, observations, and comments. The comments helped us to improve our work so that it can be easily accessible to readers in the scientific community.

Comment 1. In the “Introduction” part, it is suggested for the authors to give more information on the novelty and significance of this manuscript.

Authors’ Response:Thanks for highlighting this critical point. The novelty and significance of this manuscript were highlighted in the introduction and abstract of the revised version.

Comment 2. It is suggested for the authors to add a scheme to present the synthesis of PUF and the PUFS, as well as the adsorption of CPF.

Authors’ Response: Thanks for the valuable input to the manuscript. Your suggestions were added and a scheme (1) to present the synthesis of PUF and the PUFS, as well as the adsorption of CPF was created. The  added scheme can be seen in line 127 in the Materials and Methods section, as marked in yellow in the revised manuscript.

Comment 3. In Figure 8a and 9b, the error bars of data points should be added.

Authors’ Response:Thank you so much for addressing this point. We added the error bars in Figures 8a and 9b according to your comment and highlighted in the revised version of the manuscript in lines 467 and 503, respectively.

Comment 4. To prove the advantages of the formed PUFS for removing CPF by comparing with other materials, the authors should add the comparison on the adsorption performance of all materials.

Authors’ Response: We agree with your comment on the importance of comparing the advantages of the synthesized PUFS for extraction CPF with other materials on the adsorption performance of CPF from aqueous solutions. Hence, we would like to indicate your attention to this comparison is present in our manuscript in Table 3 in the result and discussion section (Line: 465). We added Table 3 to display and distinguish the difference between the maximum adsorption capacity with other materials.

Finally, we also appreciate your valuable time for editing our manuscript. We hope this revised version is suitable enough for publication.

Round 2

Reviewer 1 Report

Revised has been done properly except the language editing. Please do it again. Please mention the amount of PUFS gm you prepared from 10 gm PUF. Thus, you can determine the yield (%). 

Author Response

Dear Sir,

I want to thank you for your comments. They significantly improved our work as you can see the amount of corrections in our revised manuscript. The amount of PUFS were added to the manuscript and highlighted in green. The added sentence read as:

" the average amount of synthesized PUFS is 11.86 gm, as well as its increase in the yield is 15.68%"

Thank you for the time you spent on our manuscript.

Best regards,